# Discrimination between Complete versus Non-Complete Pathologic Response to Neoadjuvant Therapy Using Ultrasensitive Mutation Analysis: A Proof-of-Concept Study in *BRCA1*-Driven Breast Cancer Patients

**DOI:** 10.3390/ijms24031870

**Published:** 2023-01-18

**Authors:** Anna P. Sokolenko, Fedor V. Moiseyenko, Aglaya G. Iyevleva, Alexandr O. Ivantsov, Georgiy D. Dolmatov, Ksenia V. Shelekhova, Elizaveta V. Gulo, Anastasya X. Topal, Elizaveta V. Artemieva, Nuriniso H. Abduloeva, Nikita A. Rysev, Daria A. Barsova, Natalia V. Levchenko, Nikita M. Volkov, Vitaliy V. Egorenkov, Vladimir M. Moiseyenko, Evgeny N. Imyanitov

**Affiliations:** 1Department of Tumor Growth Biology, N.N. Petrov Institute of Oncology, 197758 Saint Petersburg, Russia; 2Department of Medical Genetics, St.-Petersburg Pediatric Medical University, 194100 Saint Petersburg, Russia; 3City Cancer Center, 68A Leningradskaya Street, Pesochny, 197758 Saint Petersburg, Russia

**Keywords:** *BRCA1* mutation, breast cancer, neoadjuvant chemotherapy, pathologic complete response, review

## Abstract

Neoadjuvant chemotherapy (NACT) for breast cancer (BC) often results in pathologic complete response (pCR), i.e., the complete elimination of visible cancer cells. It is unclear whether the use of ultrasensitive genetic methods may still detect residual BC cells in complete responders. Breast carcinomas arising in *BRCA1* mutation carriers almost always carry alterations of the *TP53* gene thus providing an opportunity to address this question. The analysis of consecutive BC patients treated by NACT revealed a higher pCR rate in *BRCA1*-driven vs. *BRCA1*-wildtype BCs (13/24 (54%) vs. 29/192 (15%), *p* < 0.0001). Twelve pre-/post-NACT tissue pairs obtained from *BRCA1* mutation carriers were available for the study. While *TP53* mutation was identified in all chemonaive tumors, droplet digital PCR (ddPCR) analysis of the post-NACT tumor bed revealed the persistence of this alteration in all seven pCR-non-responders but in none of five pCR responders. Eleven patients provided to the study post-NACT tissue samples only; next-generation sequencing (NGS) analysis revealed mutated *TP53* copies in all six cases without pCR but in none of five instances of pCR. In total, *TP53* mutation was present in post-NACT tissues in all 13 cases without pCR, but in none of 10 patients with pCR (*p* < 0.000001). Therefore, the lack of visible tumor cells in the post-NACT tumor bed is indeed a reliable indicator of the complete elimination of transformed clones. Failure of ultrasensitive methods to identify patients with minimal residual disease among pCR responders suggests that the result of NACT is a categorical rather than continuous variable, where some patients are destined to be cured while others ultimately fail to experience tumor eradication.

## 1. Introduction

Neoadjuvant chemotherapy (NACT) is commonly utilized for the treatment of breast cancer (BC) with the aim to reduce the tumor size and, consequently, minimize the extent of surgical intervention [1]. In addition, NACT is a valuable tool for the analysis of the efficacy of various anticancer drugs: it involves treatment-naïve patients and allows the biological analysis of post-NACT tumor masses [2]. Ideally, NACT is aimed at achieving pathologic complete response (pCR), i.e., the complete elimination of residual tumor cells. Multiple studies indicate that pCR is strongly associated with being cured from BC, particularly in patients with triple-negative disease and in women with HER2-driven carcinomas [3,4,5].

Up to 10% of BC patients carry germ-line mutations in BC-predisposing genes. BC development in carriers of *BRCA1/2* pathogenic variants usually involves somatic inactivation of the remaining *BRCA1/2* allele in the tumor tissue. These malignancies are characterized by the deficiency of DNA repair by homologous recombination and are highly sensitive to platinum compounds, PARP inhibitors, mitomycin C, and some other drugs [6]. BRCA1 is essential for taxane-mediated apoptosis [7]; some, although not all, clinical studies suggest decreased efficacy of taxane-containing schemes in *BRCA1*-driven BCs [8]. There are several investigations comparing the efficacy of NACT in *BRCA1/2* mutation carriers vs. non-carriers [9,10,11,12,13,14]. The analysis of *BRCA1/2*-associated BCs is often focused on platinum-containing regimens, given that the high vulnerability of BRCA1/2-deficient cells to cisplatin and carboplatin was consistently observed in various laboratory experiments [6,8,15,16]. However, there is still no consensus regarding the optimal choice of NACT regimens for *BRCA1*- and *BRCA2*-mutated BCs; therefore, further accumulation of clinical data is required.

Biallelic inactivation of the *BRCA1* gene, which is observed in hereditary BC arising in *BRCA1* mutation carriers, is not compatible with cell viability. Hence, the pathogenesis of *BRCA1*-driven cancers almost always involves the inactivation of the *TP53* gene to prevent an apoptotic death of BRCA1-deficient cells. Some relatively easily accessible genetic techniques, particularly droplet digital PCR (ddPCR), allow the detection of single mutated gene copies in the presence of a huge excess of normal DNA. Therefore, once we know the identity of the *TP53* mutation in a given tumor, it is possible to develop an individual ddPCR assay, which will be able to detect minimal residual disease. Visual morphological analysis of postsurgical tumor tissues obviously has some limitations in sensitivity. It remains unclear whether the pathologic complete response indeed reflects the ultimate elimination of malignant cells or, alternatively, the use of an ultrasensitive method for the detection of cancer-specific mutations will identify instances of pCR accompanied by the presence of invisible residual tumor cells in surgically excised tissues. This study aimed to address this issue.

## 2. Results

### 2.1. Clinical and Pathologic Response to NACT

Among 229 consecutive BC patients who were treated by neoadjuvant chemotherapy, there were 25 *BRCA1* carriers (5382insC: *n* = 22, 185delAG: *n* = 2, 4153delA: *n* = 1) and 204 women without recurrent *BRCA1* alterations. A more detailed clinical description is presented in Appendix A.

The clinical response according to RECIST was evaluated in 208 cases. The total rate of objective clinical response was 16/19 (84%) in *BRCA1* carriers vs. 117/189 (62%) in non-carriers (*p*-value 0.08, Fisher’s exact test; Table 1).

The difference was more pronounced and reached the level of statistical significance for patients receiving platinum-containing neoadjuvant regimens. The majority of *BRCA1*-driven BCs have triple-negative receptor phenotype [17]; therefore, it is appropriate to limit the comparison to this category of BC cases. Strikingly, the trend towards a higher rate of clinical response to platinum-containing therapy in *BRCA1* mutation carriers vs. non-carriers was retained in BCs, which did not express estrogen, progesterone, and HER2 receptors (Table 1).

The rate of pCR defined as ypT0/ypN0 was 42/216 (19%) in the entire cohort. Noticeably, while the clinical response rate by RECIST was only slightly higher in *BRCA1*-associated vs. *BRCA1*-wildtype BC (Table 1), this difference was significantly more pronounced for pCR (13/24 (54%) vs. 29/192 (15%), *p*-value < 0.0001; see Table 2).

Increased pCR rates in *BRCA1* carriers were observed both in patients receiving platinum-containing therapy and in women exposed to NACT without platinum. Surprisingly, when the analysis was limited to triple-negative BCs, the difference in pCR rates tended to be numerically higher in patients treated by non-platinum NACT, while the superiority of platinum-containing regimens did not reach the threshold of statistical significance (Table 2).

### 2.2. TP53 Mutation Analysis

The results of molecular analysis of *TP53* status in primary tumors and post-NACT surgically removed tissues are presented in Table 3. Representative examples of *TP53* mutation testing are given in Figure 1.

*TP53* mutations were identified in all 12 available chemonaive tumor tissues. Strikingly, *TP53* mutation was retained in post-NACT samples in all 7 cases with visible residual tumor cells (i.e., Miller–Payne score < 5). However, despite the use of an ultra-sensitive method for detecting *TP53* alterations, mutated *TP53* copies were not observed in 5 post-NACT samples obtained from patients with pCR. Among 11 patients with post-NACT tissues only, 6 patients failed to achieve pCR; *TP53* mutation were detected in all these samples. In contrast, 5 post-NACT samples with pCR did not contain *TP53*-mutated gene copies. In total, failure to achieve pCR was observed in 13 of 23 analyzed women; *TP53* mutation was present in post-NACT tissues in all these cases. 10 patients demonstrated pCR after NACT; none of the post-NACT tissues for this group of women had detectable mutated copies of the *TP53* gene (*p* < 0.000001).

## 3. Discussion

This study demonstrates that pCR rate upon NACT is considerably higher in *BRCA1*-driven vs. sporadic BCs. Importantly, pCR detected by visual microscopic inspection of surgically excised tissues indeed reflects a complete elimination of tumor cells, as shown by the use of ultrasensitive methods of detection of minimal residual disease.

This investigation presents a real-world experience of the use of NACT in *BRCA1* mutation carriers and non-carriers in consecutive patients with BC. Small number of patients with *BRCA1*-driven BCs is a weakness of this report. Despite this limitation, the study size was sufficient to demonstrate that *BRCA1*-associated tumors are generally more chemosensitive than non-hereditary BCs, irrespective of whether the comparison is performed in the entire patient population or in women with triple-negative BC. This observation is in good agreement with similar data sets [9,10,11,12,13,14]. However, neither this investigation nor prior studies allow us to define which therapeutic scheme is the best for BC patients carrying *BRCA1* germ-line mutation. Although both preclinical and clinical data suggest high efficacy of platinum-containing regimens in this category of patients, the analysis of the real-world data is complicated because the majority of non-platinum drug combinations involve anthracyclines. The mechanisms of action of anthracyclines include the induction of DNA double-strand breaks, so in this respect, they are similar to platinum salts [7].

Pathological complete response rates in *BRCA1/2*-driven BC vary across the studies. It is of note that many reports pool together *BRCA1* and *BRCA2* mutation carriers, despite the fact that *BRCA1*-driven BCs are usually triple-negative while *BRCA2*-associated tumors are often positive for estrogen and progesterone receptors. There are only several reports which evaluated the response to NACT separately for *BRCA1*- and *BRCA2*-driven BCs (Table 4).

Interestingly, even when the analysis is limited to *BRCA1* mutation carriers, the pCR rate within this relatively homogenous BC subgroup varies between 20% and 61%, with both these extreme results obtained using apparently the same treatment regimen, e.g., cisplatin monotherapy (Table 4). It is difficult to estimate whether the subjective nature of morphological assessment contributes at least in part to these deviations. Surprisingly, there are no investigations which utilized ultrasensitive methods, such as ddPCR or NGS, for the validation of pCR results. Our experiments, which involved the search for mutated *TP53* gene copies, convincingly demonstrate that pCR is indeed an indicator of the complete elimination of tumor cells, at least in the primary tumor site. This is an important finding confirming that pCR is a truly valuable end-point for BC clinical studies [2,3,4,5].

The results of this study may provide interesting insights into the mechanisms of tumor response to therapeutic interventions. The distribution of effects of antitumor therapy in a given patient group is commonly viewed as a continuous variable, with some subjects experiencing maximal benefit at the one end, some patients showing disease progression on the opposite end, and the majority of cases demonstrating various degrees of intermediate effect. This study indicates that the response to NACT in *BRCA1*-driven BC is a categorical variable: some tumors are destined to completely disappear, with no signs of the residual disease even on the “molecular level”, while others cannot be eradicated by conventional therapeutic tools. The origin of the post-NACT residual tumor masses can be related either to the selection of pre-existing therapy-resistant tumor clones or the plasticity of malignant cells resulting in the adaptation to cytotoxic pressure. Intratumoral heterogeneity of chemonaive *BRCA1*-driven cancers has been shown in several studies [20,21]. Indeed, while the majority of cells constituting the tumor mass demonstrate the loss of the wild-type *BRCA1* allele, there is a small fraction of BRCA1-proficient malignant cells which are resistant to therapeutic pressure and undergo expansion during NACT [21]. Probably, tumors lacking the above-described intratumoral heterogeneity or having limited potential for cellular plasticity are more vulnerable to the NACT. A comprehensive comparison of tumor properties in pCR responders vs. non-responders is of high importance for future studies in this field.

## 4. Materials and Methods

The study retrospectively considered 991 patients with suspected potentially operable BC, who were consecutively referred for surgical treatment to the City Cancer Center (St.-Petersburg, Russia) within the years 2017–2018. These patients were subjected to germ-line testing for 4 recurrent *BRCA1* mutations, which were previously shown to be characteristic for Russia and some neighboring countries with predominantly a Slavic population (5382insC [c.5266dupC], 4153delA [c.4034delA], 185delAG [c.68_69delAG], 300T>G [C61G]). NACT was given to 231 of these 991 patients. Two patients were excluded from further analysis due to the absence of clinical and pathologic response data. 216 patients underwent surgery and were analyzed for the frequency of pCR, which was defined as the absence of BC cells in breast tumor bed or regional lymph nodes (ypT0/ypN0). In 162 cases, both pre- and postoperative tissue samples were evaluated, and a pathologic response score according to the Miller–Payne system was available [22] (Figure 2a).

Surgically removed tissues were available for molecular analysis in 15 patients with *BRCA1* germ-line mutations. In order to increase the sample size for the *TP53* study, we added to the investigation 8 cases of *BRCA1*-mutated BC treated by NACT in the N.N. Petrov Institute of Oncology (St.-Petersburg) within the years 2016–2018. Both pre- and post-NACT tissues were accessible for mutation testing in 12/23 cases; the analysis of the remaining 11 BCs was limited by post-NACT biological material. Residual tissues obtained from patients with pCR were dissected within the tumor bed (see an example in Figure 1).

The type of *TP53* mutation was analyzed with the targeted next-generation sequencing as described by Sokolenko et al. [23]. The individual digital droplet PCR tests with probes specific either to normal or mutant *TP53* sequences were developed for 9 out of 12 BC cases with available pre-/post-NACT tissue pairs. The description of conditions for digital droplet PCR is given in the study of Kuligina et al. [24] (Figure 1). Sequences of primers and probes are presented in Appendix A. *TP53* analysis for 11 patients with post-NACT tissues only and for 3 women with pre-/post-NACT sample doublets was limited by NGS (Figure 2b).

## Figures and Tables

**Figure 1 ijms-24-01870-f001:**
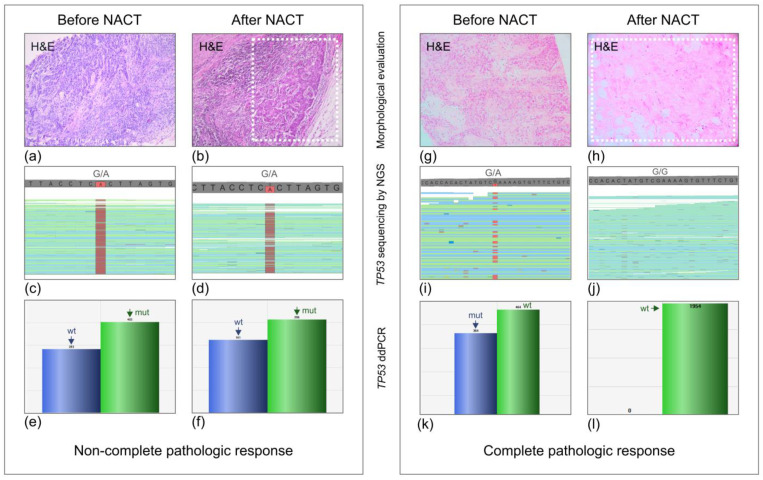
Examples of cases with non-complete (Miller–Payne score 3) and complete (Miller–Payne score 5) pathological response. Morphological and molecular data are presented. Upper panel: hematoxylin–eosin staining of primary (**a**,**g**) and residual tumors (**b**,**h**); middle panel: next-generation sequencing (NGS) reads (**c**,**d**,**i**,**j**); bottom panel: digital droplet PCR (ddPCR) droplets positive for FAM (blue) or R6G (green) (**e**,**f**,**k**,**l**).

**Figure 2 ijms-24-01870-f002:**
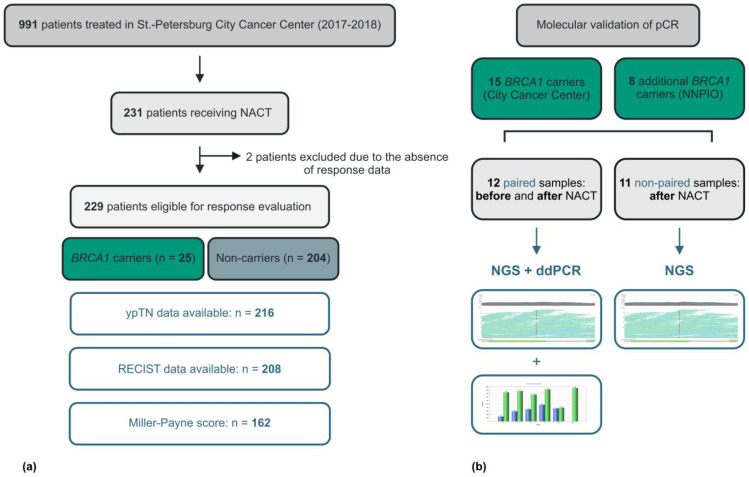
Study flow-chart. Selection of the patients (**a**). *TP53* analysis (**b**).

**Table 1 ijms-24-01870-t001:** The objective response rate in different patient groups.

	All Cases (*n* = 208)		
	*BRCA1*-carriers (*n* = 19)	Non-carriers (*n* = 189)	*p*-value
All regimens	16/19 (84%)	117/189 (62%)	0.08
Platinum-containing (*n* = 33)	12/14 (86%)	8/19 (42%)	0.02
Without platinum (*n* = 175)	4/5 (80%)	109/170 (64%)	0.66
	**Triple-negative carcinomas (*n* = 57)**		
	*BRCA1*-carriers (*n* = 18)	Non-carriers (*n* = 39)	
All regimens	15/18 (83%)	24/39 (62%)	0.13
Platinum-containing (*n* = 28)	11/13 (85%)	6/15 (40%)	0.02
Without platinum (*n* = 29)	4/5 (80%)	18/24 (75%)	1.0

**Table 2 ijms-24-01870-t002:** The pCR rate in different patient groups.

	All Cases (*n* = 216)		
	*BRCA1*-carriers (*n* = 24)	Non-carriers (*n* = 192)	*p*-value
All regimens	13/24 (54%)	29/192 (15%)	<0.0001
Platinum-containing (*n* = 37)	9/18 (50%)	3/19 (16%)	0.04
Without platinum (*n* = 179)	4/6 (67%)	26/173 (15%)	0.008
	**Triple-negative carcinomas (*n* = 59)**		
	*BRCA1*-carriers (*n* = 20)	Non-carriers (*n* = 39)	
All regimens	12/20 (60%)	6/39 (15%)	0.001
Platinum-containing (*n* = 30)	8/15 (53%)	3/15 (20%)	0.128
Without platinum (*n* = 29)	4/5 (80%)	3/24 (13%)	0.007

**Table 3 ijms-24-01870-t003:** Pathologic response according to Miller–Payne and *TP53* mutations in residual tissues.

ID	*BRCA1* Mutation	Subtype	Ki-67, Grade	NACT (No. of Cycles)	Miller–Payne Score	*TP53* Mutation	*TP53* Detected after NACT	Samples Available for Analysis (Method)
NBC1	c.5266dupC	Triple-negative	17%, G2	P (2)	2	P151R	Yes	Pre-NACT (NGS) and post-NACT (NGS)
NBC3	c.5266dupC	Luminal B	69%, G3	AC (4)	3	Ex8 55-bp del	Yes	Post-NACT (NGS)
NBC4	c.5266dupC	Triple-negative	75%, G3	TP (4)	4	R213X	Yes	Pre-NACT (NGS) and post-NACT (ddPCR)
NBC6	c.5266dupC	Triple-negative	33%, G3	P (6)	5 (pCR)	R213X	No	Pre-NACT (NGS) and post-NACT (ddPCR)
NBC8	c.5266dupC	Triple-negative	90%, G3	P (4)	3	Y163C	Yes	Pre-NACT (NGS) and post-NACT (NGS)
NBC9	c.5266dupC	Triple-negative	49%, G3	FAC (3)	3	I255N	Yes	Pre-NACT (NGS) and post-NACT (ddPCR)
NBC10	c.5266dupC	Triple-negative	29%, G3	P (4)	5 (pCR)	-	No	Post-NACT (NGS)
NBC11	c.5266dupC	Triple-negative	64%, G3	TP (4)	5 (pCR)	Y234C	No	Pre-NACT (NGS) and post-NACT (ddPCR)
NBC12	c.68_69delAG	Triple-negative	36%, G3	P (6)	1	R306X	Yes	Pre- NACT (NGS) and post-NACT (NGS)
NBC13	c.68_69delAG	Triple-negative	Nd	TP (6)	3	M237I	Yes	Pre-NACT (NGS) and post-NACT (ddPCR)
NBC14	c.5266dupC	Triple-negative	85%, G3	FAC (6)	Nd * (RCB II)	c.755delT	Yes	Post-NACT (NGS)
NBC15	c.5266dupC	Triple-negative	75%, G3	TP (2)	5 (pCR)	-	No	Post-NACT (NGS)
NBC16	c.5266dupC	Triple-negative	60%, G3	TP (4)	5 (pCR)	-	No	Post-NACT (NGS)
NBC18	c.5266dupC	Triple-negative	65%, G3	AC (4)	5 (pCR)	-	No	Post-NACT (NGS)
NBC19	c.5266dupC	Triple-negative	75%, G3	TP (5)	5 (pCR)	Y220C	No	Pre-NACT (NGS) and post-NACT (ddPCR)
NBC21	c.5266dupC	Triple-negative	90%, G3	AC (4), T (12)	Nd * (RCB II)	c.473insTCCG	Yes	Post-NACT (NGS)
NBC23	c.5266dupC	Triple-negative	80%, G3	FAC (4), T (12)	5 (pCR)	-	No	Post-NACT (NGS)
NBC24	c.5266dupC	Luminal B	30%, G2	FAC (4), P (9)	4	R273L	Yes	Pre-NACT (NGS) and post-NACT (ddPCR)
NBC25	c.5266dupC	Triple-negative	80%, G3	FAC (4), P (12)	5 (pCR)	c.445delT	No	Pre-NACT (NGS) and post-NACT (ddPCR)
NBC26	c.5266dupC	Triple-negative	90%, G3	FAC (4), P (12)	5 (pCR)	R273H	No	Pre-NACT (NGS) and post-NACT (ddPCR)
NBC29	c.5266dupC	Triple-negative	90%, G3	AC (4), T (12)	2	Ex7 17-bp del	Yes	Post-NACT (NGS)
NBC31	c.4034delA	Triple-negative	90%, G3	AC (4), P (6)	1	C275Y	Yes	Post-NACT (NGS)
NBC32	c.4034delA	Triple-negative	90%, G3	AC (4), P (2)	2	R273C	Yes	Post-NACT (NGS)

Abbreviations: AC, anthracycline and taxane; ddPCR, digital droplet PCR; FAC, 5-fluoruracil, anthracycline, and cyclophosphamide; NACT, neoadjuvant chemotherapy; NGS, next-generation sequencing; P, platinum; RCB, residual cancer burden; T, taxane; TA, taxane and anthracycline; TAC, taxane, anthracycline, and cyclophosphamide; TP, taxane and platinum. * In two cases where the Miller–Payne score could not be determined due to the absence of primary tumor, the amount of residual tumor cells was estimated by RCB score.

**Table 4 ijms-24-01870-t004:** Major studies evaluating pCR rate in BC patients with *BRCA1* germ-line mutations.

Study	No. of *BRCA1/2* Carriers	NACT Composition	pCR Rate
Arun et al. [18]	57 *BRCA1* and 23 *BRCA2* carriers	Three treatment groups: anthracycline-taxane (AT) therapy, anthracycline-based regimens without a taxane, or single-agent taxane	The highest pCR rates were seen in AT group in all patient categories: 46% in *BRCA1* carriers, 13% in *BRCA2* carriers, and 22% in non-carriers
Byrski et al. [8]	107 *BRCA1* carriers	Cisplatin 75 mg/m^2^ every 3 weeks for 4 cycles	65/107 patients (61%)
Pfeifer et al. [19]	19 *BRCA1* carriers	Anthracycline-based schemes without taxanes and taxane-containing regimens	The overall pCR rate was 32%; pCR was achieved in 56% of patients receiving anthracycline-based regimens without taxanes
Paluch-Shimon et al. [9]	34 *BRCA1* carriers	4 cycles of dose-dense doxorubicin (60 mg/m^2^) and cyclophosphamide (600 mg/m^2^) every 2 weeks followed by dose-dense paclitaxel (175 mg/m^2^) every 2 weeks for 4 cycles or 12 weekly cycles of paclitaxel (80/m^2^)	68% in *BRCA1* carriers compared to 37% in non-carriers (*p* = 0.01)
Bignon et al. [11]	46 *BRCA1* carriers, 6 *BRCA2* carriers, and one *BRCA1/BRCA2* double heterozygote; all triple-negative BCs	Anthracycline-based schemes either with taxanes or without taxanes	The overall pCR rate in *BRCA1* mutation carriers was 38%; it was statistically non-significantly higher in patients receiving NACT with taxanes compared to patients who didn’t receive taxanes (12/25 (43%) vs. 5/18 (28%), respectively)
Wunderle et al. [15]	43 *BRCA1* carriers, 16 *BRCA2* carriers	Either anthracycline-based therapy (4 cycles of epirubicin (80–90 mg/m^2^) and cyclophosphamide (600 mg/m^2^) every 3 weeks, followed by 12 cycles of paclitaxel (80–90 mg/m^2^) weekly) or platinum-based therapy (6 cycles of carboplatin AUC5 on day 1 and paclitaxel (80–90 mg/m^2^) on days 1, 8, and 15, every 3 weeks)	pCR rate was 58% for *BRCA1* mutation carriers, 44% for *BRCA2* carriers, and 23% for sporadic carcinomas. In *BRCA1* carriers, pCR rate was 11/25 (44%) among patients receiving anthracycline-based therapy and 14/18 (78%) in patients receiving platinum-based therapy (*p* = 0.03)
Fasching et al. [13]	74 *BRCA1* carriers and 16 *BRCA2* carriers, all triple-negative BCs	4 cycles of epirubicin/cyclophosphamide followed by 4 cycles of docetaxel; or 4 cycles of epirubicin/cyclophosphamide/bevacizumab followed by 4 cycles of docetaxel/bevacizumab	pCR rate was 49% in *BRCA1* carriers and 56% in *BRCA2* carriers
Tung et al. [16]	81 *BRCA1* carriers, 35 *BRCA2* carriers, and two double heterozygotes *BRCA1/BRCA2*	Cisplatin 75 mg/m^2^, 4 cycles of AC (doxorubicin 60 mg/m^2^ and cyclophosphamide 600 mg/m^2^), 4 cycles every 2 (dose-dense) or 3 weeks	pCR in the cisplatin arm: *BRCA1:* 20%; *BRCA2:* 13%; pCR in the AC arm: *BRCA1:* 28%; *BRCA2:* 25%

## Data Availability

The data that support the findings of this study are available from the corresponding author upon reasonable request.

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
