# Peer review of "Discrimination between Complete versus Non-Complete Pathologic Response to Neoadjuvant Therapy Using Ultrasensitive Mutation Analysis: A Proof-of-Concept Study in BRCA1-Driven Breast Cancer Patients"

_ijms, 2023, doi:10.3390/ijms24031870_

Round 1

Reviewer 1 Report

The authors evaluated in the present study the pathologic response to NACT among BRCA1 positive breast cancer patients and non-positive breast cancer patients. The study is of scientific importance and deserves publication; however, the following issues should be addressed:

1) The introduction should clearly state the purpose of the study as well as the targeted population. It should also explain its need, given that this evidence has been previously presented.

2) The methods section should indicate this IRB approval number. It should also state the actual methodology that was used: prospective retrospective, cohort or case control etc

3) Inclusion and exclusion criteria (ex stage, histology, age, other comorbidities, NACT discontinuation for toxicity or other causes, number of NACT cycles etc) should be stated as these may influence the outcomes of interest.

4) The methods section should not include findings of the study; hence, Figure 1 should be placed in the results, along with the comment. The methodology section should be clear and thorough. Some incomplete phrases are seen (ex "The type of TP53 mutation was analyzed with the targeted next-generation sequenc-203 ing as described in [23].").

5) The discussion is inappropriately structured. The auhtors should read the paper of Doherty et al (BMJ. 1999 May 8;318(7193):1224-5.doi: 10.1136/bmj.318.7193.1224.) and provide the following information: findings of their study, comparison to existing literature, limitations and strengths, implications for physicians and future researchers

Author Response

Comment: The introduction should clearly state the purpose of the study as well as the targeted population. It should also explain its need, given that this evidence has been previously presented.

Response: We now state in the Introduction: “It remains unclear whether the pathologic complete response indeed reflects the ultimate elimination of malignant cells or, alternatively, the use of an ultrasensitive method of detection of cancer-specific mutations will identify instances of pCR accompanied by the presence of invisible residual tumor cells in surgically excised tissues. This study aimed to address this issue.”

Comment: The methods section should indicate this IRB approval number. It should also state the actual methodology that was used: prospective retrospective, cohort, or case control etc.

Response: This information is now provided in the Section Institutional Review Board Statement: “The study was approved by the local Ethics Committee (No. 20/25, January 23, 2020).”  We now clarify in the Materials and Methods section that this is a retrospective study.

Comment:  Inclusion and exclusion criteria (ex stage, histology, age, other comorbidities, NACT discontinuation for toxicity or other causes, number of NACT cycles etc.) should be stated as these may influence the outcomes of interest.

Response: The study flow is now given in the Figure 2. Clinical information for patients receiving NACT is described in Table S1 (Supplementary Material).

Comment: The methods section should not include findings of the study; hence, Figure 1 should be placed in the results, along with the comment. The methodology section should be clear and thorough. Some incomplete phrases are seen (ex "The type of TP53 mutation was analyzed with the targeted next-generation sequencing as described in [23].").

Response: Figure 1, accompanied by the appropriate reference in the text, has been moved to the Results section. We re-phased the sentences which looked incomplete.

Comment: The discussion is inappropriately structured. The authors should read the paper of Doherty et al (BMJ. 1999 May 8;318(7193):1224-5.doi: 10.1136/bmj.318.7193.1224.) and provide the following information: findings of their study, comparison to existing literature, limitations and strengths, implications for physicians and future researchers

Response: Thank you very much for pointing our attention to the paper of Doherty et al., 1999! We have incorporated brief description of the main findings in the beginning of the section, emphasized the limitations of the study, and extended the discussion with the focus on significance of our investigation.  

Reviewer 2 Report

In this manuscript, the authors compared complete to non-complete pathologic response (pCR) using droplet digital PCR (ddPCR) and next-generation sequencing (NGS). Using those methods, they attempted to detect residual breast cancer cells in complete responders. NGS and ddPCR revealed that TP53 mutations exist in pCR-non-responders but not in pCR-responders. Therefore, they found that the lack of visible tumor cells in the post-neoadjuvant chemotherapy (NACT) tumor bed is a reliable indicator of the complete elimination of transformed clones. The results of this manuscript only confirmed the reliability of pCR but have a clinical implication. The following points need to be addressed.

1 The data in Figure 1 yield the primary evidence for the results of this manuscript. I recommend that figure 1 move to the results section.

2 In the Introduction or the Results, an explanation of platinum-containing compounds needs to be included because in Tables 1 and 2, the results are differentiated for compounds with and without platinum. They should explain why they made the distinction.

3 The supplementary Figure 1 helps the readers understand this manuscript. I recommend that Supplementary Figure 1 is changed to Figure 1.

4 In Table 3, the authors should specify NGS and ddPCR samples.

5 Line 60: unvestigations (?)           

6 What does “NGS” stand for?

Author Response

Comment: The data in Figure 1 yield the primary evidence for the results of this manuscript. I recommend that figure 1 move to the results section.

Response: This is done.

Comment: In the Introduction or the Results, an explanation of platinum-containing compounds needs to be included because in Tables 1 and 2, the results are differentiated for compounds with and without platinum. They should explain why they made the distinction.

Response: We have inserted an appropriate explanation in the Introduction: The analysis of BRCA1/2-associated BCs is often focused on platinum-containing regimens, given that high vulnerability of BRCA1/2-deficient cells to cisplatin and carboplatin was consistently observed in various laboratory experiments [6, 8, 18, 19].

Comment: The supplementary Figure 1 helps the readers understand this manuscript. I recommend that Supplementary Figure 1 is changed to Figure 1.

Response: This is done. This figure is now Figure 2.

Comment: In Table 3, the authors should specify NGS and ddPCR samples.

Response: This is done.

Comment: Line 60: unvestigations (?)

Response: Thank you for noticing this typing error, we have corrected it.

Comment: What does “NGS” stand for?

Response: Next generation sequencing. We have inserted the explanation of this abbreviation in the Abstract and in the text.

Round 2

Reviewer 1 Report

No further comments

Reviewer 2 Report

The authors have satisfactorily addressed the points which I noted.